# Elucidation of host and symbiont contributions to peptidoglycan metabolism based on comparative genomics of eight aphid subfamilies and their *Buchnera*

Thomas E. Smith *, Yiyuan Li, Julie Perreau, Nancy A. Moran

Department of Integrative Biology, University of Texas at Austin, Austin, Texas, United States of America

* smit4227@gmail.com

## Abstract

Pea aphids (*Acyrthosiphon pisum*) are insects containing genes of bacterial origin with putative functions in peptidoglycan (PGN) metabolism. Of these, *rlpA1-5*, *amiD*, and *ldcA* are highly expressed in bacteriocytes, specialized aphid cells that harbor the obligate bacterial symbiont *Buchnera aphidicola*, required for amino acid supplementation of the host's nutrient-poor diet. Despite genome reduction associated with endosymbiosis, pea aphid *Buchnera* retains genes for the synthesis of PGN while *Buchnera* of many other aphid species partially or completely lack these genes. To explore the evolution of aphid horizontally-transferred genes (HTGs) and to elucidate how host and symbiont genes contribute to PGN production, we sequenced genomes from four deeply branching lineages, such that paired aphid and *Buchnera* genomes are now available for 17 species representing eight subfamilies. We identified all host and symbiont genes putatively involved in PGN metabolism. Phylogenetic analyses indicate that each HTG family was present in the aphid shared ancestor, but that each underwent a unique pattern of gene loss or duplication in descendant lineages. While four aphid *rlpA* gene subfamilies show no relation to symbiont PGN gene repertoire, the loss of aphid *amiD* and *ldcA* HTGs coincides with the loss of symbiont PGN metabolism genes. In particular, the coincident loss of host *amiD* and symbiont *murCEF* in tribe Aphidini, in contrast to tribe Macrosiphini, suggests either 1) functional linkage between these host and symbiont genes, or 2) Aphidini has lost functional PGN synthesis and other retained PGN pathway genes are non-functional. To test these hypotheses experimentally, we used cell-wall labeling methods involving a D-alanine probe and found that both Macrosiphini and Aphidini retain *Buchnera* PGN synthesis. Our results imply that compensatory adaptations can preserve PGN synthesis despite the loss of some genes considered essential for this pathway, highlighting the importance of the cell wall in these symbioses.

**Data Availability Statement:** Sequencing data can be found at NCBI (PRJNA758084). Genome annotations and other data generated during the

current work can be found at Zenodo (https://zenodo.org/record/5517159). Scripts and code used in this study can be found at GitHub (https://github.com/lyy005/peptidoglycan_related_genes_in_basal_aphids). All accession numbers and URLs to data repositories can be found in the main text of the manuscript.

**Funding:** This work was supported by National Institutes of Health (https://www.nih.gov/) Awards 5F32GM126706 (to T.E.S.) and 5R35GM131738 (to N.A.M), and National Science Foundation (https://www.nsf.gov/) Award 1551092 (to N.A. M.). The funders had no role in study design, data collection and analysis, decision to publish, or preparation of the manuscript.

**Competing interests:** The authors have declared that no competing interests exist.

## Author summary

Throughout evolution, animals have sometimes gained novel abilities by acquiring bacterial genes through horizontal gene transfer. For some insects harboring bacterial symbionts, horizontally-transferred genes may enable hosts to regulate symbiosis by influencing symbiont cell wall metabolism. While mealybug horizontally-transferred genes work collectively to synthesize the symbiont cell wall, the role of aphid horizontally-transferred genes in symbiont cell wall metabolism is unclear. We examined whether different aphid horizontally-transferred genes co-occur with symbiont genes underlying cell wall metabolism across different aphid lineages, indicative of linked function. We included 17 aphid species representing eight distantly related lineages, four of which we sequenced for this study. We found that two of the three horizontally-acquired gene families are present only when symbionts possess cell wall pathway genes, while the third shows no correlation. These results reveal that despite their putative involvement in symbiont cell wall synthesis, aphid horizontally-acquired genes operate independently from one another and likely have lineage-specific functions. Furthermore, we observed that symbiont cell wall synthesis is maintained in one aphid lineage despite loss of genes considered essential for producing the cell wall, implying that other adaptations preserve the cell wall in aphid species with incomplete cell wall synthesis pathways.

## Introduction

Horizontal gene transfer (HGT) has played a pivotal role in eukaryotic evolution, granting novel functions to its recipients and providing the means to expand to new ecological environments [1–7]. HGT has been particularly relevant in the evolution of eukaryotes that live in close association with bacteria [8,9]. At the extreme, the majority of mitochondrial and plastid genes have been transferred to the host genome [10,11], affording hosts complete control of their bacterial symbionts and the benefits they provide. In contrast, transfer of symbiont genes to the host genome is absent or limited in a number of insect endosymbioses [12–15]. In these systems, host genomes may instead harbor genes anciently acquired from bacteria other than the current symbiont(s) that have obtained roles in maintaining extant symbioses through nutrient provisioning [15,16] or the metabolism of peptidoglycan (PGN), the building block of bacterial cell walls [12–14]. For example, the genome of the mealybug *Planococcus citri* contains horizontally-transferred genes (HTGs) encoding enzymes of the PGN synthesis pathway that perfectly complement the PGN gene repertoire of the symbionts [14] and contribute to the construction of the symbiont cell wall [17].

Aphids are sap-feeding insects that rely on their *Buchnera* symbionts to provide essential amino acids lacking in their diet. The pea aphid (*Acyrthosiphon pisum*) *Buchnera* genome is missing many genes considered to be essential in free-living relatives like *Escherichia coli* [18], including most PGN pathway genes. Pea aphids contain eight HTGs with putative functions in PGN metabolism: *ldcA*, *amiD*, *rlpA1-5*, and *bLys* [12,13], of which all but *bLys* show greatly elevated expression in bacteriocytes, the specialized host cells that harbor *Buchnera*, relative to other aphid tissues [13]. The putative functions of the remaining seven HTGs correspond to specific steps in bacterial PGN metabolsim: *amiD* and *rlpA* putatively encode an amidase and lytic translgycosylase (LT), respectively, both of which are involved in cell wall remodeling, while *ldcA* encodes a putative L,D-carboxypeptidase of the PGN recycling pathway (Fig 1). Aphids also contain invertebrate-type (i-type) lysozymes that are highly expressed in bacteriocytes[19], especially during late adulthood when symbionts and bacteriocytes are degraded

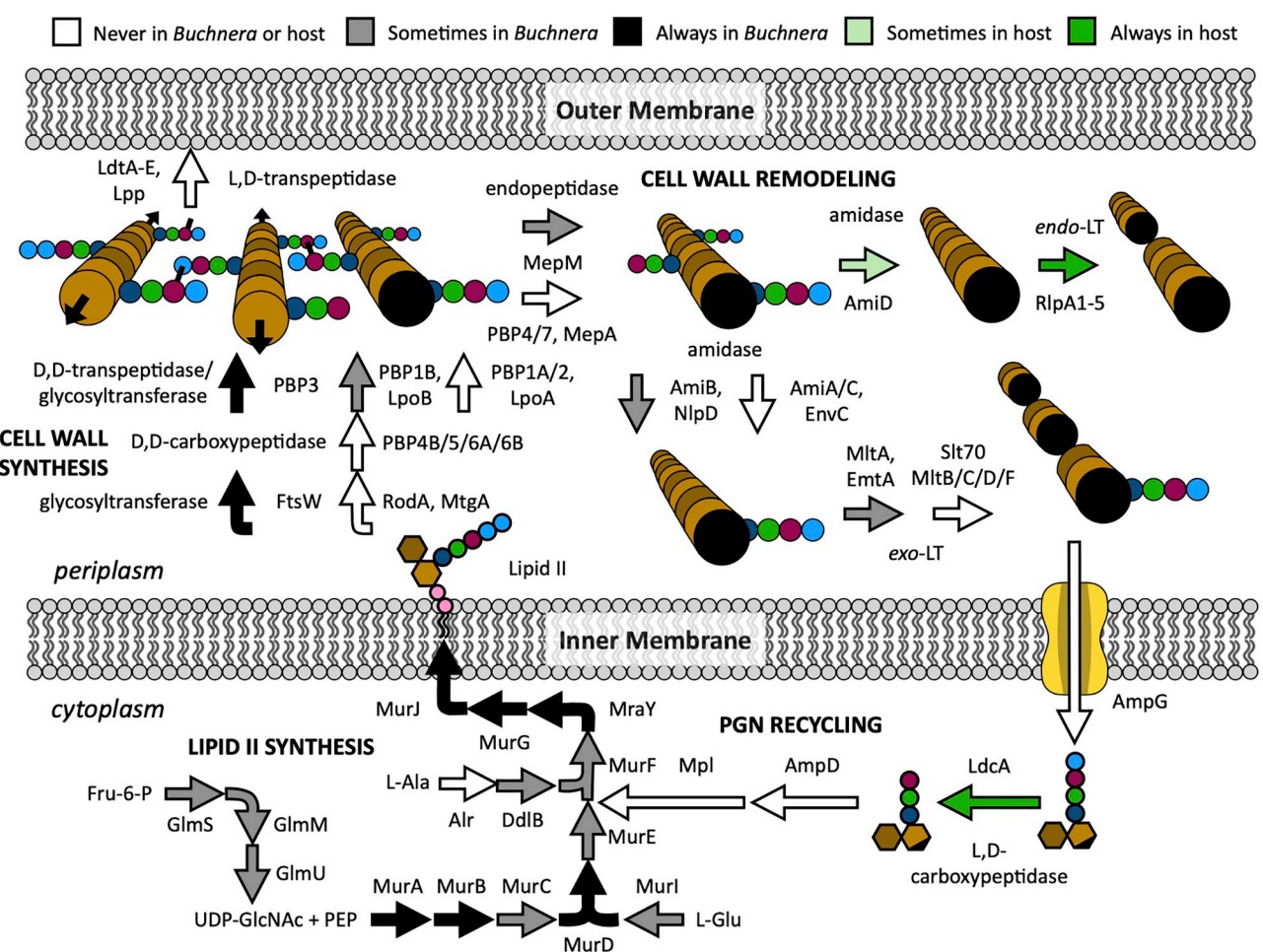

**Fig 1. Hypothetical metabolic reconstruction of *Buchnera* PGN metabolism with potential involvement of aphid HTGs in the Aphidinae subfamily (including tribes Macrosiphini and Aphidini).** PGN metabolism is divided into four pathways: PGN recycling and lipid II synthesis in the cytoplasm, and cell wall synthesis and remodeling in the periplasm. Arrows depict enzymatic reactions, with black, gray, and white colors indicating complete, sporadic, or no conservation of the responsible gene(s) among ten *Buchnera* genomes. Light and dark green arrows denote incomplete and complete conservation of host HTGs. For PGN cartoons, dark and light brown indicate GlcNAc and MurNAc glycans, respectively, while black-ended light brown indicates anhydro-MurNAc. Stem peptides consist of L-Ala (dark blue), iso-D-Glu (green), *meso*-diaminopimellic aicd (*m*Dap) (maroon), and D-Ala (light blue) amino acids. For lipid II, the phosphate groups of undecaprenyl-diphosphate are shown in pink.

[20]. Thus, aphid hosts may have the enzymatic potential to control *Buchnera* by influencing its cell wall structure.

Despite the apparent relationship between aphid HTGs and symbiont cell wall metabolism, it is unclear from their putative functions alone how they might contribute to this process. The *A. pisum amiD* and *rlpA1-5* genes may be functionally redundant with *Buchnera amiB* and *emtA* or *mltA*, respectively, and *Buchnera* completely lacks the PGN recycling pathway to which aphid LdcA putatively belongs (Fig 1) [21]. Recent evidence shows that aphid LdcA instead acts on PGN within or shed from the *Buchnera* cell wall [22]. One way to establish whether genes are functionally linked is to determine whether they are preserved or lost together throughout evolution using phylogenetic profiling—the correlated presence of genes within genomes of distinct evolutionary lineages can provide a signal of functional interactions [23,24]. Similarly, if host and symbiont genes interact, they should co-occur in paired host and symbiont genomes across diverse lineages, whereas genes that do not interact will be gained or

lost independently of one another. Such an approach has been used previously to detect host-symbiont collaborations among various mealybug species [25]. Gene distributions across species have also been used to correlate the lack of class A penicillin-binding proteins with an intraceullar lifestyle among distantly related bacteria [26]. Within the aphid superfamily Aphidoidea, *Buchnera* symbionts vary widely in genome size and content, including PGN genes [27], providing a natural source of variation to test whether host HTGs coincide with certain symbiont PGN genes or pathways. Furthermore, in a few aphid lineages, *Buchnera* has been lost and replaced with a novel symbiont type [28,29] or has been joined by additional symbionts [30,31]. While many *Buchnera* genomes are available from diverse aphid species [27,30–34], only 13 aphid genomes have been sequenced, annotated, and published to date, and most are concentrated in a single subfamily, the Aphidinae (S1 Table).

In this work, we introduce draft genomes of aphid species from four additional aphid subfamilies: *Geopemphigus sp.* (Fordinae), *Stegophylla sp.* (Phyllaphidinae), *Chaitophorus viminalis* (Chaitophorinae), and *Pemphigus obesinymphae* (Pemphiginae). We applied phylogenetic profiling across the Aphidoidea superfamily to evaluate whether certain host and symbiont PGN genes co-occur across different aphid species, supporting a potential functional relationship. For the 17 annotated aphid genomes, we assayed the incidence (presence/absence) of host genes with putative functions in PGN metabolism, including homologs of the *A. pisum* HTGs *ldcA*, *amiD*, *rlpA*, and *bLys*, as well as PGN pathway genes from the corresponding primary symbiont of each host species. For the latter, we used a manually-curated database of primarily *E. coli* and *Buchnera* genes representing known PGN genes and members of the cell division and flagellar basal body (FBB) pathways, both of which may interact with or depend on cell wall metabolism [35–37]. We show that the PGN gene repertoire of each aphid species varies for both host and symbiont. The pattern of aphid *ldcA* and especially *amiD* gene loss correlates with the absence of PGN pathway genes in their respective symbiont(s), while *rlpA* genes, though variable in the incidence and abundance of four distinct paralogous groups, are conserved among all aphids. These observations imply that aphid HTGs are not all involved in symbiont PGN metabolism and likely function independently of each other, suggesting that the encoded enzymes do not cooperate to complement a missing multi-step pathway in symbionts as proposed for mealybug HTGs [25]. Furthermore, a clear difference in gene repertoire exists between two tribes of the Aphidinae subfamily. Relative to tribe Macrosiphini, Aphidini aphids lack *amiD* and their symbionts lack 2–3 genes central to lipid II biosynthesis (*murF*, *murC*, and/or *murE*). Thus, gene repertoires suggest that Aphidini *Buchnera* cannot synthesize cell walls at all. In contrast with this hypothesis, we provide experimental evidence for functional cell wall synthesis in an Aphidini species, *Rhopalosiphum maidis*, demonstrating that 1) novel adaptations have evolved within the PGN biosynthesis pathway to preserve the cell wall in this species, and 2) genomic analysis alone is an insufficient predictor of symbiont capabilities.

## Results

### Genome sequencing and annotation

To understand the evolution of genes related to PGN pathways, we sequenced the genomes of four divergent aphid lineages, including the first genome sequences for subfamilies Fordinae (*Geopemphigus sp.*), Phyllaphidinae (*Stegophylla sp.*) and Pemphiginae (*Pemphigus sp.*), and the second genome sequence for subfamily Chaitophorinae (*Chaitophorus sp.*) after *Sipha flava* (NCBI BioProject PRJNA472250). The *Geopemphigus* genome also represents the first aphid genome for which *Buchnera* has been lost and replaced with an unrelated endosymbiont [29]. For each genome, we generated 38–58 Gb of genomic data from one short insert size Illumina library (S2 Table) and 5–7 Gb RNA-seq data (S3 Table). After assembly, estimated

genome sizes range from 245 Mb—658 Mb (S4 Table). The sequenced genomes show high completeness based on the presence of single-copy Benchmarking Universal Single-Copy Orthologs (BUSCOs) for Hemiptera (n = 2,510) [38], ranging from 97%-99% for both the assemblies (S5 Table) and our gene annotation results (S6 Table). These values are similar to those for other aphid genome sequences [33]. All sequencing data can be found at NCBI under the accession number PRJNA758084.

## Aphid HTG incidence varies across aphid species

In order to understand 1) the incidence of *A. pisum* HTG homologs among different aphid lineages and 2) the evolutionary history of host-encoded PGN genes, we reconstructed the aphid phylogeny using the four newly sequenced aphid genomes, 13 available aphid genomes, and six related insects in suborder Sternorrhyncha included as outgroups (S1 Table). The phylogeny was based on 829 single-copy genes found in at least 80% of the species (Fig 2). Genes with

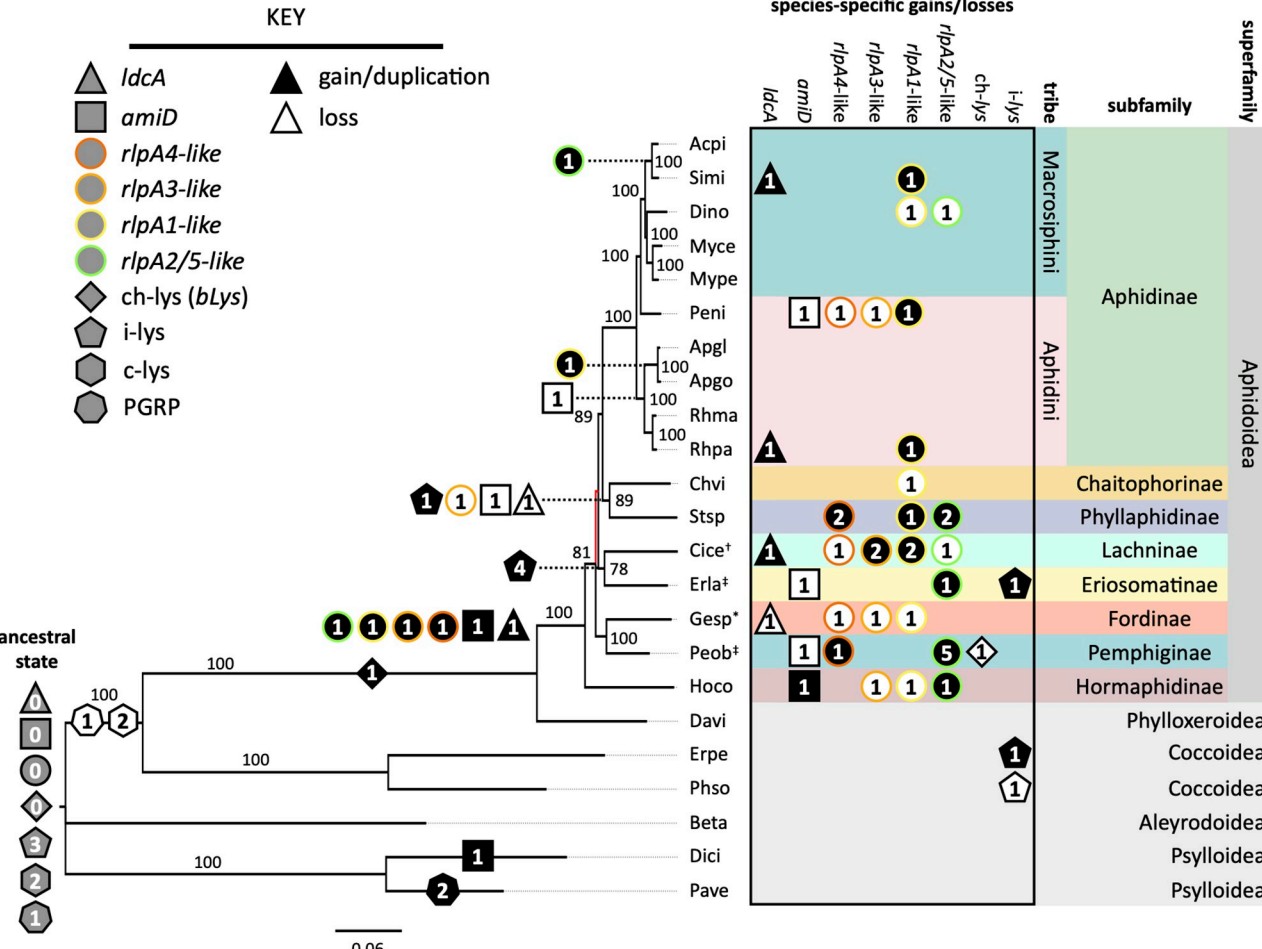

**Fig 2. Ancestral reconstruction of aphid genes that putatively interact with Gram-negative bacterial PGN.** The aphid phylogeny was constructed using maximum likelihood from a concatentated nucleotide alignment of 829 core genes shared by all 23 insect species. Bootstrap supports after 1,000 replicates are shown, with red branches indicating supports < 70%. All symbionts are *Buchnera* except those of *Geopemphigus* sp, indicated by an asterisk, which contains a novel Bacteroidetes symbiont. Host species containing known co-obligate or uncharacterized facultative symbionts are indicated with an obelisk (†) or a double obelisk (‡), respectively. Each species code represents the combined first two letters of the genus and species. The ancestral state (gray) and placement of gene gains (black), duplications (black), and losses (white) were determined using Count (38). Numbers in shapes indicate the number of gene copies involved in the event. Shapes indicate distinct gene families, and colored borders denote the four RlpA paralog types, named after the *A. pisum rlpA* genes.

putative activities pertaining to PGN metabolism, including peptidoglycan recognition proteins (PGRPs), lysozymes, and aphid HTGs (*bLys*, *ldcA*, *amiD*, and *rlpA*), were identified by BLAST search using *A. pisum* protein sequences as queries (S7 Table). With the exception of *Daktulosphaira vitifoliae* [39], the outgroup species contain bacterial symbionts, none closely related to *Buchnera*.

We first evaluated the distribution of genes encoding enzymes that target PGN and genes that represent conserved components of insect innate immunity (S7 Table). PGRPs sense bacterial infection by binding and/or cleaving PGN and activating downstream pathways that stimulate the humoral immune response, which includes increased production of bacteriostatic lysozymes [40]. Lysozymes, characterized by their *N*-acetylmuramidase activity, include members from four different glycoside hydrolase families: chicken- and invertebrate-type (c- and i-types, respectively; GH22), goose-type (g-type; GH23), viral-type (v-type; GH24), and Chalaropsis-type (ch-type; GH25) [40]. Both PGRPs and c-type lysozymes are evolutionarily conserved in insects [40,41]. We found no genes encoding PGRPs or c-type lysozymes among aphid or phylloxerid genomes, but their presence in the scale insect, whitefly [42], and psyllid [43,44] genomes support their loss in a shared ancestor of the Aphidomorpha infraorder (Fig 2). The absence of these key immune genes may have facilitated the establishment of endosymbioses in aphids [45], though their absence from phylloxerids, which lack symbionts [39] and presence in other outgroups that harbor obligate symbionts suggest that alternative forces underlie their loss from the Aphidomorpha. We observed little variation in the number of invertebrate-type (i-type) lysozymes among most insects investigated (Fig 2), indicating a functional difference between these and c-type lysozymes. Major exceptions are *Cinara cedri* (Lachninae) and *Eriosoma lanigerum* (Eriosomatinae) which harbor eight to nine i-lysozyme paralogs compared with three or four paralogs in other species. Finally, we found *bLys* in all Aphidomorpha genomes, indicating that this gene family was acquired concomitantly with the loss of PGRPs and c-type lysozymes (Fig 2). The ch-type lysozyme domain of *bLys* is most closely related to domains in prophages of *Wolbachia*-like bacteria [13,46] and is minimally expressed within bacteriocytes, implying a role outside of symbiosis [13]. The multiple independent HGTs of ch-type lysozymes to all domains of life [46] and the unique *N,O*-diacetylmuramidase activity described for some ch-type lysozymes [47,48] suggest that aphid *bLys* is more likely involved in host defense against bacterial pathogens than in symbiont PGN metabolism.

Unique to superfamily Aphidoidea are the HTGs *rlpA*, *ldcA*, and *amiD*, all of which were acquired in a shared ancestor of this superfamily (Fig 2). With the exception of an *amiD* homolog within the *D. citri* genome, none of the bacteriocyte-expressed HTGs found in aphids were detected outside of the Aphidoidea (Fig 2). In the case of AmiD, phylogenetic analysis revealed that the *D. citri* homolog clusters separately from the aphid copies, indicating that *D. citri amiD* was acquired from a bacterial source independently of aphid *amiD* (S1 Fig).

Aphid lineages vary in whether HTGs have been lost or duplicated (Fig 2). RlpA subfamilies are maintained to different extents among aphid lineages, with each genome containing at least one type (Fig 2). We examined the phylogenetic relationships of aphid RlpA proteins and found that each sequence falls within one of four paralogous subfamilies: RlpA4-like, RlpA3-like, RlpA1-like, and RlpA2/5-like (S2 Fig). RlpA4-like proteins are the most basally branching subfamily and RlpA2/5 is the most recent branch, supporting the initial analysis of *A. pisum rlpA* genes [13]. Ancestral reconstruction indicates that the most recent aphid common ancestor had all *rlpA* subfamilies, implying that the underlying gene duplications occurred in a common ancestor of the Aphidoidea (Fig 2). In contrast, *amiD* and *ldcA* have been lost independently five and two times, respectively (Fig 2). The repeated loss of these genes provides an opportunity to test our hypothesis—if aphid HTGs contribute to symbiont PGN

metabolism, the loss of aphid HTGs should correlate with the absence of interacting symbiont genes.

## Aphid *amiD* and *ldcA* coincide with the loss of symbiont PGN

We anticipated that the symbionts of host species lacking *amiD* and/or *ldcA* genes might display greater loss of genes from PGN-related pathways due to a functional link between these host genes and symbiont cell wall metabolism. To test this idea, we looked for patterns of coincidence between host and symbiont PGN gene repertoires among all available sequenced aphid species. First, we measured the incidence of 138 bacterial genes, collectively involved in PGN metabolism or two potentially related pathways, within the primary symbiont genome of each aphid species (Fig 3 and S8 Table). As noted previously, *Buchnera* from aphids of tribe Macrosiphini (subfamily Aphidinae) harbor the most complete set of PGN metabolism genes among *Buchnera* species, maintaining a nearly complete lipid II synthesis pathway and the minimal enzyme functions necessary for cell wall synthesis (glycosyltransferase and D,D-transpeptidase) (Fig 3) [21,27]. Though Macrosiphini *Buchnera* generally maintain genes encoding two of the three essential activities for cell wall remodeling, an amidase (*amiB*) and at least one lytic transglycosylase (LT; *mltA*, *emtA*) gene, they often lack the gene typically encoding the third, endopeptidase activity (*mepM*) (Fig 1). In comparison, *Buchnera* of the tribe Aphidini (subfamily Aphidinae) display more frequent loss of *murF*, *murC*, and/or *murE* of the lipid II synthesis pathway. Most other subfamilies (Chaitophorinae, Phyllaphidinae, Lachninae, Eriosomatinae, and Pemphiginae) exhibit near-complete loss of all PGN pathway genes, while *Buchnera* from *Hormaphis cornu* (subfamily Hormaphidinae) represents an intermediate, maintaining a single cell wall synthesis gene (*mrcB*, encoding the bifunctional glycosyltransferase/D,D-transpeptidase PBP1B) and partial lipid II synthesis pathway (Fig 3). Notably, no aphid primary symbionts, including *Buchnera* of most aphids and *Candidatus* Skilesia alterna (referred to hereafter as *Skilesia*) of *Geopemphigus*, maintain any of the PGN recycling genes used in our searches (S8 Table).

Next, we considered whether the presence or absence of host PGN genes coincides with the degree of symbiont PGN gene loss. The ubiquity of aphid i-type *lys* and *bLys* genes in host genomes suggests no involvement of these genes with symbiont PGN metabolism (Figs 2 and 3). While the incidence of *rlpA* paralog types varies among aphid lineages, all species contain multiple *rlpA* genes, implying no relationship with symbiont gene repertoire. This is less surprising when the functional basis of RlpA's role in bacterial cell division is considered. Bacterial RlpA proteins participate in cell division largely via the N-terminal sporulation-related repeat (SPOR) domain (Pfam 05036) that binds "denuded" (stemless) PGN glycan chains that accumulate at the septum [49,50]. In contrast, aphid RlpA proteins contain only the catalytic double-ψ ß-barrel (DPBB) domain (Pfam 03330) of RlpA [12,13]. While the RlpA DPBB domain is unique among LTs for its ability to cleave denuded PGN [51,52], the absence of SPOR domains from aphid RlpA proteins suggests that these enzymes likely do not participate in symbiont cell division. Furthermore, aphid RlpA proteins contain one or more eukaryotic inhibitor cysteine-knot (ICK)/knottin domains that are unrelated to PGN metabolism [12,13], further distancing RlpA from a role in this pathway. Collectively, these observations suggest that the DPBB domain may serve an alternative function in aphid RlpA proteins.

Aphid *ldcA* exhibits a mixed distribution among host species harboring symbionts with and without PGN. The uniform conservation of *ldcA* in the Aphidinae subfamily, where *Buchnera* maintains most PGN pathway genes (Fig 3), supports a relationship between *ldcA* and symbiont PGN metabolism, at least in this lineage. Aphid *ldcA* is also present in *H. cornu*, whose *Buchnera* harbors a partial PGN pathway (Fig 3). However, aphid *ldcA* is

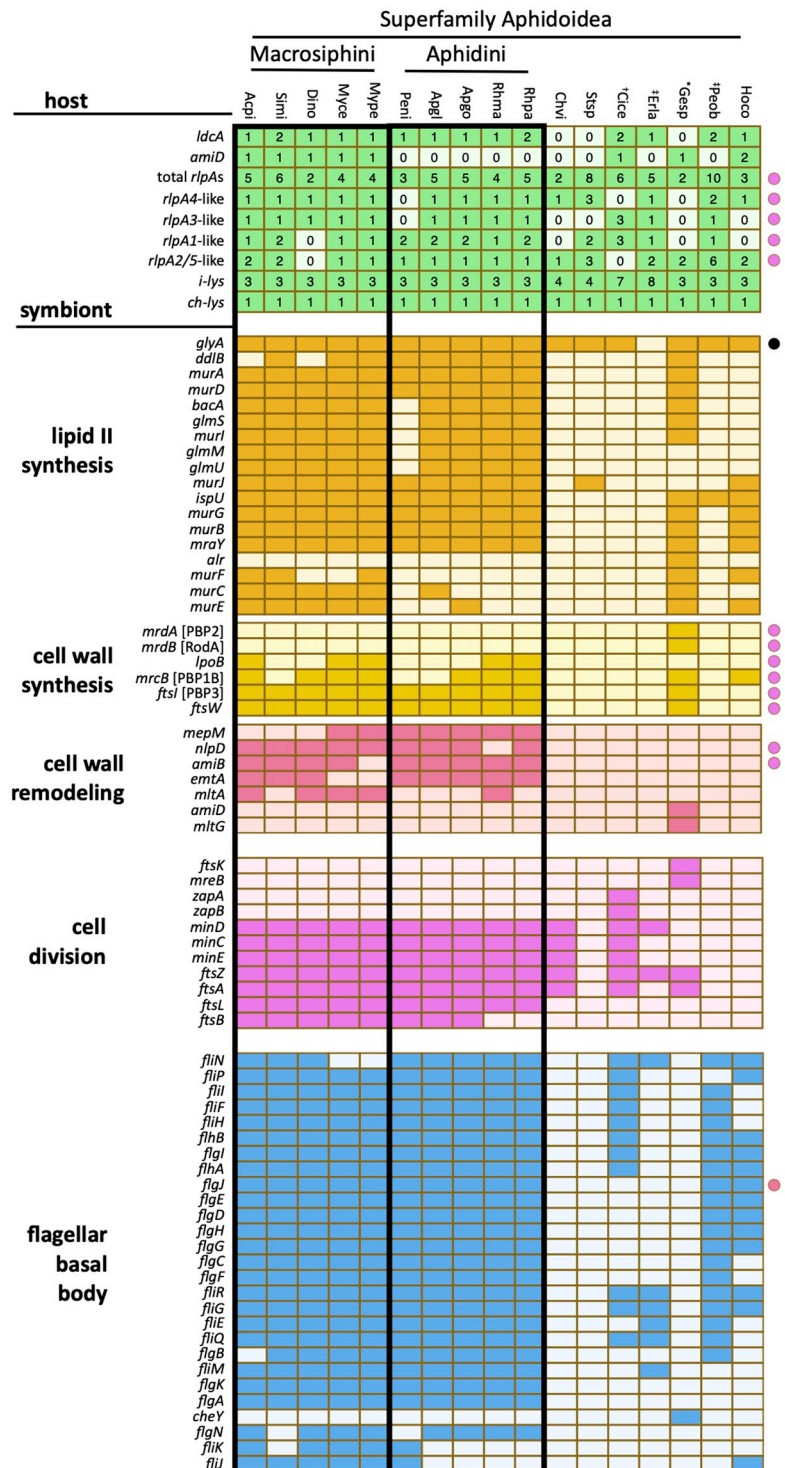

**Fig 3. Incidence of genes underlying PGN metabolism and related pathways from 17 aphid host-symbiont species.** Filled cells indicate gene presence, while empty cells indicate gene absence. For aphid genes, numbers indicate the number of gene paralogs identified per genome. For symbiont genes, filled cells all indicate a single copy per genome. All symbionts are *Buchnera* except those of *Geopemphigus* sp, indicated by an asterisk, which contains a novel Bacteroidetes symbiont. Host species containing known co-obligate or uncharacterized facultative symbionts are indicated with an obelisk (†) or a double obelisk (‡), respectively. Each species code represents the combined first two letters of the genus and species (S1 Table). Colored circles indicate genes with dual-membership in the pathway of the same color, while black indicates dual-membership in amino acid metabolism.

present in three aphid subfamilies with symbionts lacking PGN pathway genes (Eriosomatinae, Lachninae, and Pemphiginae) and is absent in two (Chaitophorinae, Phyllaphidinae). While the role of *ldcA* in the former lineages is unclear, *Buchnera* from each of the latter species has undergone significantly more gene loss relative to symbionts of other aphids [27]. Aphid *ldcA* is also absent in *Geopemphigus sp*. (Fordinae), where excessive genome reduction in *Buchnera* is thought to have led to its evolutionary replacement with the Bacteroidetes symbiont *Skilesia* [29]. Thus, our data suggest that the two independent losses of *ldcA* in these clades are associated with extensive loss of *Buchnera* genes, including most PGN genes (Figs 2 and 3).

In contrast, the incidence of aphid *amiD* in different lineages strongly suggests a link to symbiont cell wall metabolism. The loss of host *amiD* from most subfamilies with highly reduced *Buchnera* genomes (Chaitophorinae, Phyllaphidinae, Eriosomatinae, and Pemphiginae) or *Buchnera* loss (Fordinae) suggests that the gene is expendable in the absence of *Buchnera* PGN (Fig 3). The putative ability of AmiD to cleave stem peptides from anhydro-MurNAc suggests a dependence on the prior activity of LTs [53], which are distinct from lysozymes in their production of cyclic anhydro-MurNAc. As LTs are found only in *Buchnera* from Aphidinae aphids (*emtA* and/or *mltA*) and *Skilesia* (*mltG*), the presence of aphid *amiD* largely correlates with LT incidence, with the exception of *C. cedri* (Fig 3). In tribe Aphidini, the loss of host *amiD* coincides with that of *murF*, *murC*, and/or *murE* in symbionts—while the encoded enzymes putatively function in two distinct pathways of PGN metabolism (remodeling and lipid II synthesis, respectively), all four enzyme activities involve PGN stem peptides (Fig 1), implying the possibility of a functional relationship. Alternatively, a lineage-specific functional redundancy of host *amiD* with symbiont *amiB* could have rendered *amiD* obsolete. The presence of both aphid *amiD* and *ldcA* in *H. cornu* coincides with incomplete PGN gene loss in its symbiont (Fig 3), suggesting a stronger reliance on host HTGs for symbiont PGN remodeling. Finally, though *Buchnera* from *Cinara cedri* (Lachninae) lacks all PGN pathway genes, the host maintains both *amiD* and *ldcA*. Many Lachninae aphids harbor secondary symbionts that have taken on essential roles for host survival [54–56]—aphid PGN genes may regulate PGN metabolism in these secondary symbionts instead of *Buchnera*.

We also evaluated the symbiont gene repertoire of pathways related to but outside of PGN metabolism. Cell division is closely coordinated with cell wall synthesis and remodeling [34], while assembly of flagella basal bodies FBBs involves interaction with the cell wall via the ß-*N*-acetylgucosaminidase activity of FlgJ [35,36]. Like the PGN pathway, loss of both cell division and FBB genes is more extensive in all subfamilies relative to the Aphidinae (Fig 3). However, there is no apparent coincidence in these losses with those of PGN pathway genes that would implicate a dependence of either pathway on the cell wall. For example, though symbionts of both *C. viminalis* (Chaitophorinae) and *Stegophylla sp*. (Phyllaphididae) lack virtually all PGN genes, the former maintains many cell division genes, similar to the tick-borne pathogens of the genus *Ehrlichia* [21], while the latter lacks all of them (Fig 3), a unique phenomenon among obligately intracellular bacteria [21,26]. Similarly, *Buchnera* of *C. cedri*, *E. lanigerum* (Eriosomatinae), and *P. obesinymphae* (Pemphiginae) lack the PGN pathway but maintain many FBB genes (Fig 3). Furthermore, cell wall synthesis and remodeling genes that are known components of the *E. coli* divisome are not preferentially maintained in aphid symbionts that lack the PGN pathway but maintain cell division genes (Fig 3) [35]. Additionally, all *Buchnera* FlgJ enzymes lack the PGN hydrolase domain, effectively demonstrating a decoupling of FBB assembly to cell wall metabolism. Thus, the *Buchnera* cell wall generally does not appear to support cell division or FBB assembly.

## Aphidini *ldcA* genes are influenced by positive selection

Aphid HTGs present in species harboring symbionts undergoing gene loss in the PGN pathway or lacking it altogether may have evolved alternative functions through natural selection. Because *ldcA* is present in species with varying degrees of PGN pathway degradation, we were able to test which conditions might lead to positive selection. The most complete PGN pathways are found in tribe Macrosiphini, followed by tribe Aphidini—PGN pathways are near-absent in subfamily Hormaphidinae and entirely absent in all other subfamilies with *Buchnera*. We tested for lineage-specific diversifying selection in the aphid *ldcA* gene family using BUSTED [57], selecting different sets of test lineages based on the criteria described above. For example, we asked whether diversifying selection has influenced *ldcA* in members of the tribe Aphidini, where the loss of aphid *amiD* and *Buchnera murF*, *murC*, and/or *murE* may signal the absence or modification of stem peptides from *Buchnera* PGN in this lineage, leading to a potential shift in LdcA function, which also putatively targets stem peptides. We found evidence for positive selection only when all lineages or Aphidini-specific *ldcA* genes were selected as the test group (Table 1), suggesting that host *ldcA* is undergoing adaptive evolution in at least one member of the Aphidini. These results are especially interesting in the light of our recent biochemical characterization of *A. pisum* LdcA, which shows that the aphid enzyme is multifunctional and plays a novel role in PGN degradation relative to its homolog in *E. coli* [22]. In contrast, we did not detect evidence for positive selection in *ldcA* when test groups included: all Macrosiphini species, species with symbionts lacking PGN genes and lacking secondary symbionts (*P. obesinymphae* and *E. lanigerum*), species with intermediate PGN gene loss (*H. cornu*), and species in which *ldcA* has been duplicated (*Sitobion miscanithi*, *Rhopalosiphum padi*, *C. cedri*, and *P. obesinymphae*) (Table 1). Additionally, we were unable to detect evidence for positive selection of *amiD* when all genes or only duplicated genes were selected as test groups (Table 1). Thus, tribe Aphidini stands out in showing evidence of positive selection for *ldcA*. However, the lack of evidence for positive selection in some of our BUSTED tests may reflect lack of power due to small sample sizes.

## Both Macrosiphini and Aphidini *Buchnera* build cell walls

The observation that *Buchnera* of most Aphidini aphids lack *murF*, *murC*, and/or *murE* (Fig 3), representing three critical steps in lipid II biosynthesis (Fig 1), raises the question of whether these symbionts are capable of synthesizing a cell wall at all [26]. The loss of these particular genes may be coincidental, reflecting the breakdown of cell wall synthesis altogether rather than evidence of functional linkage. To test this hypothesis, we developed a method to fluorescently label *Buchnera* PGN via incorporation of ethynyl-D-alanine (EDA), a D-alanine

**Table 1. BUSTED tests for diversifying selection among aphid *ldcA* and *amiD* gene families (18 and 9 protein sequences, respectively).**

| gene family | test group | test group size | p-value |
|---|---|---|---|
| *ldcA* | all genes, all species | 18 | 0.001 |
| *ldcA* | tribe Macrosiphini | 6 | 0.499 |
| *ldcA* | tribe Aphidini | 6 | 0.000 |
| *ldcA* | subfamily Hormaphidinae | 1 | 0.104 |
| *ldcA* | symbionts without PGN | 3 | 0.148 |
| *ldcA* | genes with duplications | 8 | 0.193 |
| *amiD* | all genes, all species | 9 | 0.071 |
| *amiD* | genes with duplications | 2 | 0.057 |

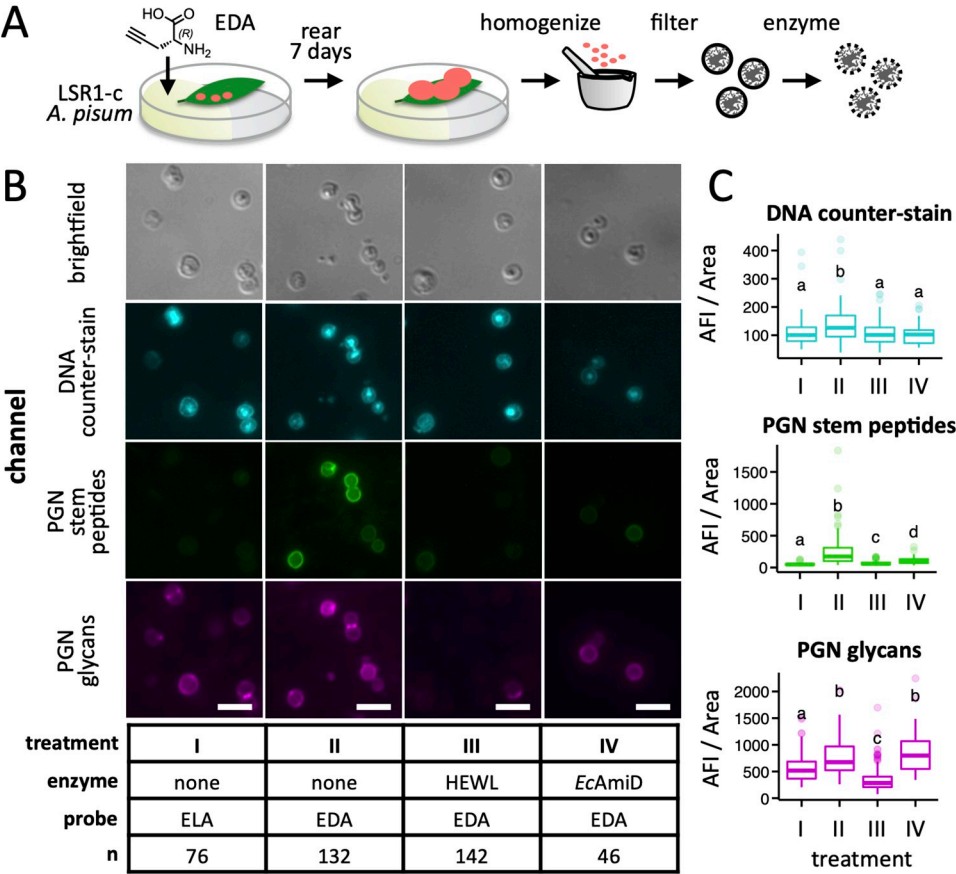

**Fig 4. Labeling of *A. pisum* LSR1-c *Buchnera* cells with EDA by host feeding.** A) Aphids were reared on plant leaves immobilized in agar containing ethynyl-D-Ala (EDA) or the control L enantiomer from birth until the 4th instar larval stage, after which *Buchnera* cells were purified from aphid homogenate. B) Cells were fixed and stained with azide-linked Alexa Fluor 488 to label PGN stem peptides (PGN stem peptides), wheat germ agglutinin CF-640R conjugate to label the PGN glycan backbone (PGN glycans), and the counter-stain DAPI to label DNA (DNA counter-stain). Hen eggwhite lysozyme (HEWL; PGN glycan-targeting) and *E. coli* AmiD (PGN stem peptide-targeting) were used to demonstrate incorporation of EDA within *Buchnera* PGN stem peptides. Images were falsely colored to aid visualization. The white scale bar represents 5 μm. C) For each combination of enzyme and probe, the average fluorescence intensity (AFI) per area was measured for each of n *Buchnera* cells using ImageJ software (S9 Table). Significant differences between means are indicated by compact letter displays above each boxplot—means with no letters in common are significantly different by the Dunn's Multiple Comparison test (p-value < 0.01 following adjustment for false discovery).

analogue compatible with click chemistry [58,59], by adapting a host feeding approach previously used to label symbiont PGN in the mealybug system [17]. Aphids were reared from birth on individual leaves immobilized in agarose gel containing EDA until reaching seven days of age (Fig 4A). To validate our approach, *Buchnera* cells were isolated from pea aphids (tribe Macrosiphini) reared on EDA, fixed, and stained with azide-linked Alexa Fluor 488 (AF488) via click reaction. *Buchnera* derived from hosts fed EDA exhibited green fluorescence at the cell periphery while control aphids raised on ethynyl-L-alanine (ELA) did not (Fig 4B), indicating 1) transport of EDA from host plant xylem to phloem, and 2) selective incorporation of EDA from plant phloem into *Buchnera* cell walls. Furthermore, the observed fluorescence was definitively linked to *Buchnera* cell wall stem peptides by treating cells with PGN-degrading enzymes prior to fixation and counter-staining with the PGN backbone-specific probe wheat germ agglutinin CF-640R conjugate (WGA640). Hen eggwhite lysozyme (HEWL) and *E. coli*

AmiD target distinct PGN features: the backbone and stem peptides, respectively. We observed a decrease in fluorescence of both EDA-AF488 and WGA640 following HEWL treatment and of EDA-AF488 alone for AmiD (Fig 4B and 4C and S9 Table), firmly associating AF488 fluorescence with PGN stem peptides.

Next, we applied the same cell wall labeling strategy to *R. maidis* aphids (tribe Aphidini) and pea aphids either cured of or containing the facultative symbiont *Serratia symbiotica*. Symbiont identities were confirmed using diagnostic FISH microscopy alongside the clickable cell wall probes (Fig 5). As expected, no cell wall fluorescence was observed in symbionts fed ELA. However, *Buchnera* cell walls were labeled with EDA in both aphid species (Fig 5A), demonstrating that even species lacking aphid *amiD* and *Buchnera murC*, *murE*, and *murF* can synthesize a cell wall. Additionally, pea aphids harboring *Serratia* show only minor incorporation of EDA into *Serratia* symbionts relative to *Buchnera* (Fig 5B), suggesting that *Serratia* differs from *Buchnera* in their primary source of D-Ala.

## Discussion

The HTGs *rlpA*, *ldcA*, and *amiD* of *A. pisum* encode putative PGN hydrolases that are more highly expressed in bacteriocytes than other host tissues, suggesting that they might play a role in *Buchnera* cell wall metabolism [13]. The putative functions of these HTGs collectively represent two of the three (LT and amidase) enzyme activities required for cell wall remodeling [60]. Furthermore, some LdcA homologs from intracellular bacteria are known to exhibit the third (endopeptidase) activity [61–63], implying that aphid HTGs might function cooperatively. Indeed, we recently showed that the *A. pisum* LdcA enzyme exhibits enhanced endopeptidase activity relative to its *E. coli* homolog, cleaving both a greater range and amount of PGN substrates [22]. Further, we found evidence of this activity in the composition of *Buchnera* PGN from pea aphids, demonstrating its biological relevance.

We examined the idea that aphid HTGs and *Buchnera* PGN metabolism might be functionally linked by identifying correlations between host and symbiont PGN gene repertoires based on newly sequenced genomes from basally branching aphid lineages. Aphid HTGs vary in their incidence and copy number among different aphid species, with each gene family displaying a unique distribution. While aphid *rlpA* genes are ubiquitous in aphids, the absence of aphid *ldcA* and *amiD* genes in some lineages coincides with greater loss of symbiont PGN pathway genes, suggesting functional linkage of *ldcA* and *amiD* with symbiont PGN metabolism. These results demonstrate that each aphid HTG family has an independent evolutionary trajectory, implying that they are unlikely to function cooperatively as mealybug HTGs do for symbiont PGN synthesis [17].

Despite the differences in the incidence of HTGs among aphids, the distribution of *rlpA*, *amiD*, and *ldcA* throughout the Aphidioidea superfamily and their absence from closely related Sternorrhynchan insects (Fig 2) suggests that their acquisition was a milestone in aphid evolution. The *Buchnera* symbiont of the most recent aphid ancestor is predicted to have had functional lipid II, cell wall synthesis, and cell wall remodeling pathways, despite evidence for significant genome reduction [27]. Thus, host HTGs could have provided an immediate benefit following their acquisition by influencing symbiont PGN metabolism (Fig 1). Aphid *ldcA* and *amiD* may continue to serve this purpose in species for which symbiont PGN pathways are maintained (Aphidinae, Hormaphidinae). Outside of these subfamilies, most symbiont PGN genes were lost, along with many other non-essential genes, due to the elevated genetic drift characteristic of endosymbionts (Fig 3) [64]. Following the loss of PGN metabolism in symbionts, HTGs were not needed and therefore lost (*amiD* in Chaitophorinae, Phyllaphidinae, Eriosomatinae, and Pemphiginae, and *ldcA* in the first two), repurposed for interaction

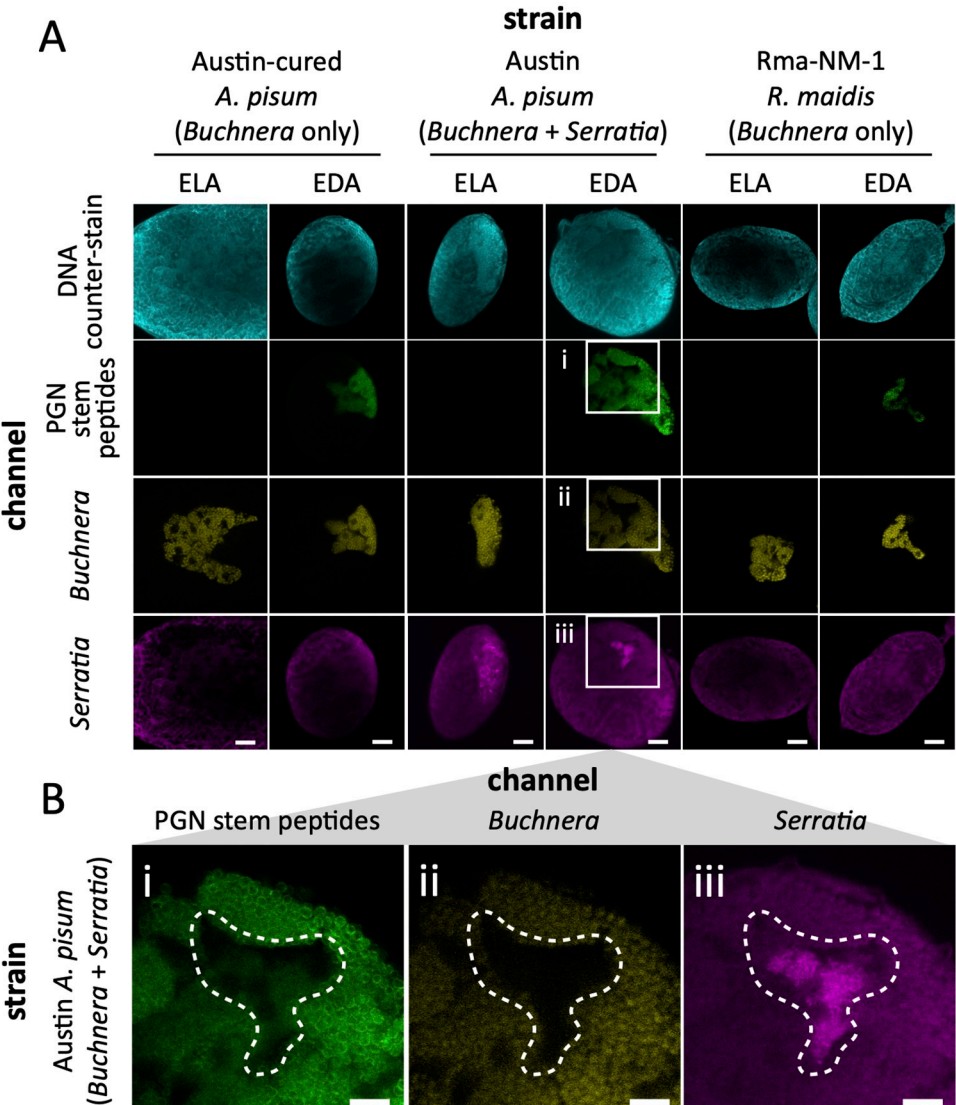

**Fig 5. FISH and cell wall labeling of Macrosiphini and Aphidini aphid symbionts.** A) Pea aphids, both cured (Austin-c; left column) and uncured (Austin; middle column) of secondary *Serratia* symbionts, and *R. maidis* (Rma-NM-1; right column) were fed ELA (left-side) or EDA (right-side) via their host plants as described in Fig 4A. Embryos removed from 4th instar larvae were fixed, treated with azide-linked Alexa Fluor 488 (AF488), hybridized with *Buchnera*- and *Serratia*-specific Cy5- and Cy3-labeled riboprobes, respectively (S10 Table), and counter-stained with DAPI. B) Enlarged cutout images (white boxes in A) of EDA-labeled, uncured *A. pisum* show overlapping AF488 fluorescence (i) with *Buchnera* cells (ii) and, to a minor extent, *Serratia* cells (iii). A bacteriocyte containing *Serratia* is highlighted with a dashed line in each image. The white scale bars represent 20 μm. Images were falsely colored to aid visualization.

with non-*Buchnera* symbionts (Lachninae, Fordinae), or maintained by purifying selection (*ldcA* in Eriosomatinae and Pemphiginae). These cases are discussed further below.

In the Lachninae aphids and in *Geopemphigus* sp., aphid *ldcA* and *amiD* may contribute to PGN metabolism in symbionts other than *Buchnera*. In *Geopemphigus* sp., *Buchnera* has been completely replaced with *Skilesia* [29]. It is possible that the *Geopemphigus amiD* gene contributes to PGN metabolism in *Skilesia*, while the *ldcA* gene could have been lost concomitantly with loss of *Buchnera* by the *Geopemphigus* ancestor. In contrast, *C. cedri* harbors both

*Buchnera* and *Serratia* [54]. The *Buchnera* symbionts of Lachninae aphids exhibit significant genome reduction relative to other subfamilies [54–56], including the loss of PGN pathway genes [31]. The gain of a co-obligate symbiont before the complete loss of *Buchnera* peptidoglycan may have prevented the loss of either aphid *ldcA* or *amiD*. Host HTGs may contribute to unconfirmed secondary or co-obligate symbionts in other species harboring *Buchnera* strains lacking the PGN pathway (Eriosomatinae and Pemphiginae). Reads from *Serratia* and *Pantoea* bacteria were found among the *E. lanigerum* and *P. obesinymphae* sequencing data, respectively [33]. Though both genera include many insect-associated bacteria [65,66], a mutualistic relationship between host and bacteria cannot be inferred from sequencing alone. Alternatively, host HTGs in lineages that lack symbiont PGN may have evolved functions that differ from their putative original roles in symbiont PGN metabolism (Eriosomatinae and Pemphiginae). Although we did not find evidence of diversifying selection for *ldcA* in these lineages (Table 1), their presence in some species lacking symbiont PGN and absence in others suggests that they remain functional.

Gene duplication is often accompanied by selection for divergence in function, although we did not detect evidence for positive selection associated with gene duplication in *ldcA* (Table 1). Phylogenetic analysis of aphid RlpA proteins reveals four groups of paralogs, each distributed throughout all aphid lineages (S2 Fig). Each paralog group exhibits a distinct phylogenetic profile and none coincides with the loss of symbiont PGN pathway genes, implying 1) distinctive functions of each paralog group, and 2) a lack of functional interaction between RlpAs and PGN metabolism. Because each aphid *rlpA* gene has eukaryotic domains with putative functions unrelated to PGN metabolism, the aphid *rlpA* gene family may have never been involved in symbiont cell wall metabolism. Knottin domains, characterized by interwoven ß-sheets stabilized by disulfide bonds, represent the conserved scaffold of a diverse range of target-selective proteins found in multiple eukaryotic kingdoms, many of which are components of venom [67]. Most knottins, including Btk1-4 in whiteflies, are encoded as small, single-domain proteins [68]. Aphids do not encode any knottin domains outside of RlpA, suggesting that the RlpA DPBB domain could have been horizontally transferred into the coding sequence of an ancestral, single-domain knottin gene. The role of knottin-RlpA proteins in the aphid-*Buchnera* symbiosis may derive from the function of their knottin domains, which are involved in immunity in whiteflies [69–71], instead of from the DPBB domain.

The question remains as to why the *Buchnera* cell wall is maintained in some species and not others. We show that cell division and FBB assembly genes, both of which interact with the cell wall in *E. coli* [35,37], are not coincident with the PGN pathway in *Buchnera* and are thus unlikely to be functionally co-dependent (Fig 3). Additionally, many intracellular environments are isotonic with the symbiont cytoplasm, eliminating the dependence on the cell wall to maintain turgor pressure [21]. Furthermore, eukaryotes typically recognize PGN via PGRPs [40], which all aphids lack (Fig 2) [45]. Our results suggest that the retention of the *Buchnera* cell wall is related to the presence of aphid *amiD* and *ldcA*—species that are able influence symbiont PGN metabolism are likely better able to control or support their symbionts, maximizing the benefits and reducing the costs associated with symbiosis. In many aphid lineages, genomic degradation in *Buchnera* has resulted in the loss of the cell wall, and this loss may eliminate the ability of hosts to regulate symbionts through the actions of AmiD and LdcA. Under this scenario, the selective forces maintaining aphid *amiD* and *ldcA* are dependent on the presence of a cell wall in symbionts, and the function of the symbiont cell wall is to enable hosts to control and/or support symbionts.

In further support of this hypothesis, we observed cell walls that fully envelope *Buchnera* cells in *R. maidis* (Fig 5) despite predictions based on gene repertoires that cell wall synthesis is not possible (Fig 3) [26]. Our experiments demonstrate the functionality of the remaining

PGN pathway genes. Thus, the concomitant loss of *Buchnera murF*, *murC*, and *murE* and host *amiD*, implies these genes are functionally related. As *Buchnera* genes underwent elimination due to drift, AmiD became useless and was subsequently lost. In Aphidini, *ldcA* genes may have enhanced importance for regulating symbiosis, as Aphidini *ldcA* could represent the sole host modulator of *Buchnera* PGN. The preservation of the cell wall despite loss of multiple genes related to the PGN stem peptide also implies the existence of compensatory mechanisms for stem peptide synthesis. The stem peptide composition of Aphidini *Buchnera* PGN may differ from that of typical Gram-negative bacteria, prompting evolutionary changes in LdcA substrate selectivity (Table 1). Furthermore, the only amino acid ligase remaining to Aphidini species is MurF, suggesting that this enzyme may have expanded its role to ligate a wide range of amino acids together to construct the stem peptide. Multifunctional enzymes likely represent a widespread adaption to genome reduction in bacteria [72,73], and two such enzymes have been described in pea aphid *Buchnera* [22,74]. However, Aphidini *Buchnera* species maintain most genes necessary to synthesize or incorporate each of the canonical stem peptide amino acids: *murI* (missing only in *Pentalonia nigonervosa*) produces D-Glu, *ddlB* synthesizes D-Ala-D-Ala, *ftsI* encodes PBP3, which forms crosslinkages involving *m*Dap, and *mepM* is an endopeptidase that cleaves those crosslinkages (Fig 3). Thus, it is unclear whether PGN structure differs between Aphidini *Buchnera* and other Gram-negative bacteria. Experimentation is needed to test conclusions drawn from genetics alone.

To date, hypotheses about aphid HTG function are informed by studies pertaining to a single aphid species, *A. pisum* (tribe Macrosiphini). In this system, HTGs are expressed more highly in bacteriocytes than in other tissues [13]. Further, RNAi knockdown [75] and correlations of host-symbiont gene expression [76] both suggest that *amiD* and *ldcA* expression positively influence *Buchnera* abundance. Finally, RlpA4 is localized to *Buchnera* [77]. Studies utilizing different aphid species, especially from tribe Aphidini and subfamily Hormaphidinae, which harbor symbionts with partial PGN pathways, are needed to establish whether the host-symbiont interactions observed for *A. pisum* are conserved in other species, especially when the potential for species-specific adaptations is high. Furthermore, sequencing of aphid-symbiont genomes from additional species within each subfamily would enable the use of statistical tools for detecting patterns in gene repertoire, whereas our analysis was limited because only a single representative species is available for most subfamilies. For example, our conclusions about the differences between the Macrosiphini and Aphidini, for which genomes are available from multiple species, are more robust than those pertaining to more basally branching but less deeply sampled lineages.

In conclusion, cell wall structure and HTG function appear to vary even among closely related aphid species, such that the relationship between aphid HTGs and symbiont PGN metabolism appears to be unique for each species. Furthermore, we emphasize the importance of experimental validation of genomics-based hypotheses, as comparative genomics alone is insufficient to predict the functional capabilities of aphids and their symbionts.

## Materials and methods

### Aphid collection

Aphids were collected from multiple locations between June, 2020 and May, 2021 (S1 Table). Aphids were removed from leaves or galls, flash frozen in liquid nitrogen, and stored at -80°C. Excess wax was removed from *Geopemphigus* aphids by dipping individuals in Dawn Professional Grade Liquid Detergent (0.1% v/v), then rinsing with deionized water. Multiple individuals regardless of developmental stage were pooled per sample, with each *Geopemphigus* and *Pemphigus* sample derived from a single gall, and thus expected to contain members of a single

all female clonal lineage. *Geopemphigus* sp. is an undescribed species corresponding to the "straight galls" found on *Pistacia texana* [29].

## Sequencing and annotation

Genomic DNA was extracted from aphid samples using the DNeasy Blood & Tissue Kit (Qiagen) following the manufacturers' protocol for DNA purification from insects using a morar and pestle and adding RNase cocktail (5 μl; Invitrogen) along with proteinase K. DNA concentration was measured using the Qubit dsDNA BR Assay Kit (ThermoFisher). To improve the assembly and gene annotation, we also extracted RNA from the four species. For *Geopemphigus sp.* and *Stegophylla* sp. samples, RNA was extracted using the RNeasy Plus Mini Kit (Qiagen) following the modifications described by Smith and Moran [75]. Genomic DNA was removed from RNA samples using the RQ1 RNase-free DNase Kit (Promega) and repurified using the RNeasy MiniElute Cleanup Kit (Qiagen). For *P. obesinymphae* and *C. viminalis* samples, RNA was extracted using the Quick-RNA MiniPrep Kit (Zymo Research) according to the manufacturers' protocol. RNA concentration was measured using the Qubit ssRNA BR Assay Kit (ThermoFisher). Whole genome DNA (350 bp insert size) and poly-A enrichment mRNA libraries were prepared and sequenced on an Illumina NovaSeq 6000 instrument (2x150) by Novogene Corporation Inc. For *Geopemphigus sp.*, we downloaded the already available genome sequencing data via NCBI under NCBI SRA accession number SRR6849201 [29]. For *H. cornu*, we manually translated the available nucleotide sequences of all protein-coding genes (S1 Table).

Our bioinformatics workflow is summarized in S3 Fig. First, we removed low quality sequences and sequencing adaptors from raw reads for both DNA and RNA sequencing data using Trimmomatic version 0.38 [78]. We first assembled the genomes with reads from DNA sequencing data using SPADES version 3.14.1 [79] with kmer sizes (-k) = 21, 33, 55, 77, 99, 127. Then we further improved the genome continuity by using the paired-end information from RNA-seq data. We mapped RNA-seq data to SPADES scaffolds using HISAT2 version 2.2.1 [80] and scaffolded the SPADES assemblies using P_RNA_scaffolder [81] with default settings.

To remove potential contamination and retrieve symbiont genomes, we first aligned scaffolds to NCBI NT database (last accessed January 27, 2021) using BLASTN megablast with "-culling_limit 5 -evalue 1e-25" in BLAST+ version 2.11.0 [82]. Then we mapped DNA sequencing data to the assemblies to estimate sequencing depth using bwa version 0.7.17 [83]. We created blobplots based on the BLAST results and BAM files using BlobTools version 1.1.1 [84]. Sequences that were annotated as human or bacterial were removed from downstream analysis. Sequences that were annotated as *Buchnera*, *Bacteroidetes*, *Rhizobiales*, *Rickettsiales*, or *Pantoea* were extracted as the symbiont genomes.

Lastly, as we detected potential contaminations from *Pemphigus* and ants in the *Chaitophorus* genome assembly (reflecting the origin of these samples from field sites where these organisms coexisted), we aligned *Chaitophorus* scaffolds to *Pemphigus* scaffolds using BLAST with e-value = 1e-10. *Chaitophorus* scaffolds were removed if 1) the scaffolds were ≥ 1000 bp and have at least 1000 bp matches with 95% identity to *Pemphigus* scaffolds, or 2) the scaffolds were < 1000bp and showed at least 95% identity with any length mapped to *Pemphigus* scaffolds. Scaffolds assigned as Hymenoptera by BlobTools were also removed from downstream analysis. We evaluated the completeness of genomes using BUSCO version 4.1.4 [38] with the Hemiptera gene set (n = 2,510).

For gene annotation, we first soft-masked the assemblies using Tantan [85] in Funannotate version 1.8.8 [86]. We used BRAKER2 version 2.1.5 [87,88] with both RNA evidence and

protein evidence from closely related species (—etpmode). For RNA evidence, we mapped RNA-seq data to assemblies using HISAT2. For protein evidence, we downloaded all annotated proteins from aphid genomes on RefSeq and the gene annotation of *E. lanigerum* [33], as it is the closest species available. Lastly, we evaluated the quality of gene annotations using BUSCO Hemiptera gene sets.

### Host phylogenetic analysis

To construct the host phylogeny, we first extracted the longest isoform for each annotated gene across 23 species. We assigned genes to orthologous groups using OrthoFinder version 2.5.2 [89] with default parameters. Orthologous groups exist in at least 80% of the species were used for phylogeny. For each orthologous group, we aligned amino acid sequences using MAFFT version 7.407 L-INS-i algorithm [90] and removed gappy regions using Gblocks version 0.91b [91] with default parameters. Lastly, we concatenated the resulting alignments and constructed phylogenetic trees using IQ-Tree version 2.0.6 [92] with 1,000 ultrafast bootstrap replicates. *Bemisia tabaci*, *Diaphorina citri*, and *Pachypsylla venusta* were used as outgroups.

Insect PGN pathway genes were identified using BLASTP against a custom database of insect PGN genes, including aphid HTGs. To search for *bLys* genes, the ch-type lysozyme domain of *A. pisum* bLys was used as a query in a TBLASTN search of all insect genomes. For each search with the exception of RlpA, hits were aligned using MUSCLE [93]. For RlpA, hits from the search of each *A. pisum* paralog were pooled, manually dereplicated, aligned with MUSCLE, and then manually adjusted based largely on the position of Cys residues in the ICK domains. Alignments were then trimmed using trimAl [94]. Gene trees were constructed for AmiD and RlpA protein sequence alignments using IQ-Tree 2 [92] with Transfer Bootstrap Estimates [95], with VTR4 and VTR5 models selected for *amiD* (S1 Fig) and *rlpA* (S2 Fig) gene families using ModelFinder [96]. Gene incidence was determined by counting the number of sequences per species in each alignment following manual inspection of alignment quality and sequence integrity.

We constructed a gain-loss-duplication model to explore the origin and fate of PGN pathway genes, including HTGs, lysozymes, and PGRPs, among aphids and close relatives. Using Count [97], we first performed rate optimizations for gene gain, loss, and duplication using flexible gain-loss and duplication-loss ratios for all lineages and a Poisson distribution at the tree root. Next, we used Dollo parsimony to determine the positions of gene gains and losses within the aphid phylogeny. Fig 2 incorporates the results of the reconstruction for each gene family with the exception of *rlpA3-* and *rlpA1*-like genes—since *H. cornu* branches at the base of Aphidoidea, parsimony maps the acquisition of these genes after the divergence of *H. cornu* from other aphids, but a phylogenetic analysis of RlpA protein sequences places RlpA3- and RlpA1-like paralogs as basal to RlpA2/5-like paralogs, suggesting that *H. cornu* lost the ancestral *rlpA3-* and *rlpA1*-like genes following the acquisition of *rlpA2/5*-like genes by gene duplication (S2 Fig).

### Analyses of *Buchnera* genomes

Sixty-nine bacterial genomes, including 51 *Buchnera*, 11 *S. symbiotica*, *Skilesia*, and five outgroup genomes, were used to identify orthologous gene groups belonging to the PGN and related pathways (S11 Table). As the assembled *Buchnera* genome sequence of *Sitobion miscanthi* was not available, we obtained its genome sequencing data from NCBI SRA (accession number: SRX5767526) [98], cleaned reads using Trimmomatic and assembled the *Buchnera* genome using SPADES under the same parameters described above. All other genomes are available online (S11 Table). First, all genomes were annotated using DFAST version 1.2.13

[99]. Coding sequences containing internal stop codons or frameshifts, likely representing pseudogenes, were removed from annotation files using a custom bash script (available at https://github.com/lyy005/peptidoglycan_related_genes_in_basal_aphids). Second, orthologous groups of DFAST-annotated bacterial proteins were determined using OrthoFinder version 2.5.2 [89]. The inflation parameter was set at 10 to account for the high phylogenetic distance among selected bacterial species. Seven proteomes downloaded from EnsemblBacteria, including *Bacteriodes fragilis* YCH46 (ASM992v1), *Escherichia coli* MG1655 (ASM584v2), *Buchnera* from *A. pisum* (ASM960v1), *B. pistaciae* (ASM772v1), and *Schizaphis graminum* (ASM736v1), and *S. symbiotica* from *A. pisum* str. Tucson (ASM18648v1), were included to facilitate retrieval of the gene names represented by each orthogroup. A SmartTable containing all *E. coli* MG1655 gene names and synonyms was downloaded from EcoCyc.org [100] and a custom R script, written with R version 3.6.1 [101] and operated in RStudio version 1.1.463 [102], was used to 1) ensure uniform gene nomenclature between *E. coli* and *Buchnera*, 2) identify the incidence of 138 genes involved in PGN metabolism or related pathways (S8 Table), and 3) count the number of genes per orthogroup per species (available at https://github.com/lyy005/peptidoglycan_related_genes_in_basal_aphids). The heatmaps shown in Fig 3 were constructed with the pheatmap package [103] using the same R script.

### Test of positive selection

To test whether certain aphid HTGs are under positive selection, we prepared codon alignments of each HTG protein family that match the protein alignments described above using PAL2NAL [104]. We then employed the BUSTED method (57) to test for positive selection through the Datamonkey web server [105].

### Aphid rearing for symbiont cell wall labeling

Clonal lines of *A. pisum*, including the parental lines LSR1-c and Austin-c, previously cured of secondary symbionts (Austin-cured) [106], as well as Austin aphids uncured of secondary symbionts (Austin), were reared on broad bean (*Vicia faba*) at 20˚C constant temperature with a 14L/10D daily light cycle. The clonal *R. maidis* line Rma-NM-1, collected from *Sorghum halepense* in Austin, Texas in 2017, was reared on barley (*Hordeum vulgare*) under the same temperature and light conditions as *A. pisum*. To incorporate D-alanine probes into *Buchnera* cell walls, aphids of either species were reared on leaves of their respective host plants immersed in a probe-agar solution. Individual leaves were immobilized in liquid 1.5% agar containing 35.4 mM of either EDA (D-proargylglycine; Alfa Aesar) or ELA probe (L-propargyl-glycine; Santa Cruz Biotechnology) within petri dishes (50 x 10 mm) and incubated for 24 hours. Two adult aphids ($\geq$ 10 days old) were placed on each leaf and allowed to reproduce for 24 hours. Adults were removed and offspring reared for 3–4 days on the same leaves before transferring aphids to fresh probe-fed leaves and rearing for another 3–4 days. Seven-day-old aphids were used for all downstream experiments.

### Purification of cell wall-labeled *Buchnera*

Seven-day-old *A. pisum* nymphs from six leaves per treatment were transferred to 1.5 ml microfuge tubes, surface-sterilized in bleach (500 µl; 0.5%) for 1 minute, rinsed twice in deionized water (500 µl), and homogenized in buffer A (300 µl; 25 mM KCl, 35 mM Tris, 10 mM MgCl$_2$, 250 mM sucrose, 250 mM EDTA, pH = 7.5) by mortar and pestle. Homogenates were passed through Sterile Swinnex filter holders (13 mm, EMD Millipore) loaded with UV-irradiated 100 µm nylon mesh filters (EMD Millipore) cut into 1/2 inch diameter circles using a

leather hollow punch. Mortar tubes were rinsed with additional buffer A (300 μl) and the rinses filtered and pooled with the initial filtrates. Pooled filtrates were centrifuged for 20 min at 4000 x g, 25˚C, and the pellets resuspended in buffer A and filtered again using a 5 μm nylon mesh filters. Filtrates were centrifuged the pellets washed with enzyme reaction buffer (300 μl; 0.1 M NaCl, 25 mM Tris, pH 7.5). Cells were centrifuged once more, the pellets resuspended in enzyme solution (50 μl; 1 μM enzyme, 0.1 M NaCl, 25 mM Tris, pH 7.5), and cell suspensions incubated at 37˚C overnight in a thermocycler with the lid set to 60˚C. Recombinant *E. coli* AmiD was recombinantly expressed (see below) while HEWL was purchased (EMD). Treated cells were centrifuged and washed twice with phosphate-buffered saline (PBS; 300 μl).

To fix and stain *Buchnera* cells, fixative (60 μl; 16% paraformaldehyde, 0.05% glutaraldehyde) was prepared and aliquoted to fresh tubes, to which sodium phosphate was added (12μl; 1 M, pH = 7.4) immediately followed by cell suspension (300 μl). The mixture was inverted twice and incubated at room temperature for 15 min, then on ice for 30 min. Fixed cells were centrifuged at 10,000 x g for 10 min, rinsed twice with PBS (300 μl), permeabilized in PBS + 0.5% Triton X-100 (300 μl) for five min, and rinsed twice more with PBS. Permeabilized cells were blocked with PBS containing 3% bovine serum albumin (BSA; 300 μl) at room temperature for 1 hour, then centrifuged. Cells were reacted with AF488 azide using the Click-iT Cell Reaction Buffer kit in 200 μl volumes of reaction mixture according to the manufacturer's specifications (Invitrogen). After washing reacted cells with BSA, cells were suspended in PBS (300 μl) and counter-stained with WGA640R (25 μg/ml) and DAPI (10 μg/ml) for 15 min in darkness. Stained cells were washed once more with PBS and suspended in 10 μl PBS, then mounted onto glass microscope slides and coverslips sealed using clear nail polish. Microscopy images were acquired using a Nikon Eclipse TE2000-U inverted microscope equipped with a Hamamatsu ORCA-Flash4.0 V2 camera and Elements software (Nikon; version 4.50.00). Cells were visualized under brightfield (100 ms), GFP ($\lambda_{ex}$ = 470 nm, $\lambda_{em}$ = 535 nm, exposure time = 2 s), Cy5 ($\lambda_{ex}$ = 640 nm, $\lambda_{em}$ = 700 nm, exposure time = 500 ms), and DAPI ($\lambda_{ex}$ = 395 nm, $\lambda_{em}$ = 460 nm, exposure time = 2 s) channels. Fluorescence images were falsely colored green, magenta, and cyan for these channels, respectively. Several multi-channel images were recorded for each sample.

To quantify the effects of enzyme treatment of cellular fluorescence of each fluorescent probe, images were analyzed using ImageJ software [107]. For each image, regions of interest (ROIs) were established around individual cells visualized in the DAPI channel using the freehand selection tool. The area and mean fluorescence across each of the four channels was calculated for each ROI using the "Multi Measure" function within the ROI Manager tool and exported as csv files. Measurements for each image were uploaded to RStudio [102] and analyzed using custom R scripts written with tidyverse package functions [108]. The mean fluorescence intensity per area was calculated for the GFP, Cy5, and DAPI channels, and statistical comparisons made for each using the dunn.test package (S9 Table) [109], which performs a Kruskal-Wallis test followed by the Dunn test to extract significant pairwise differences (p-value < 0.01 after correction for flase-discovery rate). Plots were generated using ggplot2 [110] and compact letter displays added using the cldList function from the rcompanion package [111]. Custom scripts are available at GitHub (https://github.com/lyy005/peptidoglycan_related_genes_in_basal_aphids).

## Cell-wall labeling and FISH microscopy in aphid embryos

FISH microscopy largely followed the methods of Koga *et al.* [112]. Embryos were dissected from several seven-day-old, probe-fed Austin *A. pisum* and *R. maidis* aphids in buffer A and

temporarily stored in 70% ethanol before fixing in Carnoy's solution (5 ml; ethanol:chloroform:acetic acid at a 6:3:1 ratio) for ≥ 30 min at room temperature. Fixed embryos were then washed once with 70% ethanol (5 ml) and thrice with 100% ethanol (5 ml) for 5 min per wash with gentle agitation, then bleached overnight in 1% $H_2O_2$ in ethanol. Bleached embryos were then washed thrice with 100% ethanol and stored at -20°C in ethanol. For hybridization with FISH probes, embryos were washed thrice with PBST (PBS + 0.2% Tween-20; 5 ml each), twice with hybridization buffer (20 mM Tris-HCl (pH 8.0) 0.9 M NaCl, 0.01% SDS, 30% v/v deionized formamide; 5 ml each), then incubated in hybridization buffer containing FISH probes (100 nM per Cy5-ApisP2A and Cy3-PASSisR 16S rRNA probe (S10 Table), 20 ng/ml DAPI; 3 ml) in darkness at room temperature overnight with gentle agitation. For the click reaction, hybridized embryos were washed twice with PBS + 3% BSA (1 ml), then blocked in PBS + 3% BSA (1 ml) for 60 min at room temperature with gentle agitation before replacing the solution with Click-iT reaction containing AF488 azide (500 μl) as described previously. Reacted embryos were washed twice with PBS (1 ml) and gentle agitation, then mounted onto glass depression slides with a drop of SlowFade Diamond Antifade Mountant (Molecular Probes) and fixed with coverslips sealed with clear nail polish. Images in Fig 5 were recorded on a Zeiss LSM 710 confocal microscope.

## Construction of *E. coli* AmiD protein expression vector

The *E. coli amiD* gene was cloned into the pET-28b bacterial expression vector as an N-terminally His-tagged coding sequence identical to that used by Uehara *et al.* [53]. *E. coli* DH5α genomic DNA was purified using the DNeasy Blood and Tissue kit following the protocol for Gram-negative bacteria (Qiagen). PCR amplification of *amiD* was performed with Phusion DNA polymerase (NEB), primers 1 and 2 (S12 Table), 1:200 volumes of genomic DNA template per 10 μl reaction, and the following thermocycler program: an initial denaturation (98°C, 1 min) followed by 35 cycles of denaturation (98°C, 15s), annealing (61–66°C, 15 s), and elongation (72°C, 35s) and a final elongation step (72°C, 5 min). Amplified DNA was purified using the QIAquick PCR Purification kit (Qiagen), and digested with NdeI-HF and XhoI restriction enzymes (NEB). The pET-28b vector was digested with the same restriction enzymes and dephorphorylated using Antarctic Dephosphorylase (NEB). Digested *amiD* insert was ligated into vector by inclubating with T4 DNA ligase (NEB) at 4°C overnight. The ligation reaction was transformed into chemically competent DH5α cells, plated on LB agar plates with kanamycin (50 μg/ml), and grown at 37°C overnight. Colonies were screened for the expected insert size by colony PCR using Taq Polymerase (ThermoFisher) with primers 3 and 4 (S12 Table) and the following thermocycler program: an initial denaturation (94°C, 3 min) followed by 30 cycles of denaturation (94°C, 30s), annealing (46°C, 30s), and elongation (72°C, 80s), and a final elongation step (72°C, 7 min). Positive colonies were grown in LB liquid culture (10 ml) plus kanamycin (50 μg/ml) at 37°C, 220 rpm overnight and plasmids purified using the QIAprep Spin Miniprep kit (Qiagen). Purified plasmids were verified by Sanger sequencing using primers 3 and 4 (S12 Table).

## Production and purification of *E. coli* AmiD enzyme

Chemically competent *E. coli* Rosetta (DE3) cells were transformed with 1 μl of purified plasmid, plated on LB agar plates with kanamycin (50 μg/ml) and chloramphenicol (25 μg/ml), and grown at 37°C overnight. For each strain, six colonies were used to inoculate six wells of a deep 24-well plate containing LB (6 ml) plus kanamycin (50 μg/ml) and chloramphenicol (25 μg/ml), and the plate shaken at 37°C, 220 rpm overnight. Wells were pooled and the mixture (5 ml) used to inoculate an LB culture (1L) containing kanamycin (50 μg/ml) and

chloramphenicol (25 μg/ml) in an Erlenmyer flask (2.8 L). The culture was shaken at 37˚C, 200 rpm until an $OD_{600}$ of 0.4–0.6 was reached, upon which the cultures were chilled to 18˚C, induced with isopropyl β-D-1-thioglactopyranoside (IPTG, 1 mM), and shaken overnight at 18˚C. Cells were harvested by centrifugation at 4000 rpm, 4˚C for 30 min and the pellets frozen at -80˚C until purification.

To purify proteins, cell pellets (from 500 ml culture) were thawed on ice and suspended in lysis buffer (35 ml; 0.5 M NaCl, 25 mM imidazole, 10% glycerol, pH 8.0, with freshly added 0.4 mg/ml HEWL, 1 mM *N*-methyl-*p*-toluenesulfonamide, 1 mM $MgCl_2$, and 10 μg/ml DNaseI). Cell suspensions were rocked at 4˚C for 1 hour, vortexed, and homogenized using a Emulsi-Flex-C3 high pressure homogenizer. Lysates were centrifuged at 13,000 rpm, 4˚C for 30 min and the supernatants decanted onto Ni-NTA resin (2ml; Qiagen) pre-equilibrated with lysis buffer. The lysate-resin slurry was gently mixed and the flow-through eluted. The resin was washed twice with lysis buffer (25 ml), twice with elution buffer (25 ml; 0.1 M NaCl, 5 mM HEPES, 25 mM imidazole, 10% glycerol, pH 8.0), and once with each of the following elution buffers of increasing imidazole: 50 mM, 100 mM, 200 mM, and 500 mM imidazole (5 ml). Fractions were analyzed by SDS-PAGE (S4 Fig), and those containing proteins of interest were pooled, packed into 3500 MWCO SnakeSkin Dialysis Tubing (Thermo), and dialyzed at 4˚C over the course of two days with four exchanges of dialysis buffer (1 L; 0.1 M NaCl, 25 mM Tris, 10% glycerol, pH 7.5, with 5 mM 2-mercaptoethanol added to the first two exchanges). Dialyzed proteins were concentrated using 10 kDa MWCO Amicon Ultra-15 centrifugal filters (EMD Millipore). Protein concentrations were determined from UV absorbance (λ = 280 nm) measured on a NanoDrop Lite spectrophotometer (Thermo) using fully reduced extinction coefficients calculated using the Expasy ProtParam online tool (Swiss Institute of Bioinformatics).

## Supporting information

**S1 Table. Insect species included in this study and their corresponding genome accession numbers, sources for genome annotations, and references.**
(XLSX)

**S2 Table. Sequence statistics of genome sequencing data.**
(DOCX)

**S3 Table. Sequence statistics of RNA-seq data.**
(DOCX)

**S4 Table. Statistics of genome assemblies.**
(DOCX)

**S5 Table. BUSCO analysis of genome assemblies based on the Hemiptera gene set (n = 2,510).**
(DOCX)

**S6 Table. BUSCO analysis of gene annotations based on the Hemiptera gene set (n = 2,510).**
(DOCX)

**S7 Table. Insect genes with potential roles in symbiont PGN metabolism.** Genes were identified by BLAST search using *A. pisum* or outgroup sequences as queries. Accession numbers pertain to those of published genomes or annotations, listed in S1 Table, or those published with the present article.
(XLSX)

**S8 Table. Manually-curated database of bacterial genes involved in peptidoglycan metabolism and related pathways.** Genes were identified using Orthofinder on 69 bacterial genomes. Accession numbers refer to DFAST-annotated genomes available at GitHub (https://github.com/lyy005/peptidoglycan_related_genes_in_basal_aphids).
(XLSX)

**S9 Table. Raw ImageJ data used as input to determine the average fluorescence intensity (AFI) per area measured for each of n *Buchnera* cells in Fig 4C.**
(XLSX)

**S10 Table. Fluorescent riboprobes used in FISH microscopy experiments (Fig 5).**
(DOCX)

**S11 Table. Bacterial species included in this study and their corresponding genome accession numbers and references.** Species in bold represent those that appear in Fig 3, for which the genome of the host aphid is also available. Annotations are available at Zenodo (https://doi.org/10.5281/zenodo.5484414).
(XLSX)

**S12 Table. Primers used to construct plasmid for recombinant *E. coli* AmiD expression.** Tm values were calculated for the specific DNA polymerase used for each primer set. Sequence in bold denotes restriction enzyme recognition sites, while underlined sequence highlights the target sequence used to calculate the Tm.
(DOCX)

**S1 Fig. Independent horizontal acquisition of bacterial *amiD* genes by the common ancestor of aphids and the psyllid *D. citri*.** An alignment of 256 amino acid residues was used to construct a ML tree with 100 bootstraps. TBE was used to determine bootstrap supports (88). Branches with <70% support are indicated in red. The scale bar indicates the number of substitutions per site. The tree was rooted using the *E. coli* AmpD amidase protein.
(TIF)

**S2 Fig. Phylogenetic relationships of aphid *rlpA* genes shows clustering of four distinct paralogous groups.** An alignment of 160 amino acid residues was used to construct a ML tree with 100 bootstraps. TBE was used to determine bootstrap supports (88). Branches with <70% support are indicated in red. The scale bar indicates the number of substitutions per site. The tree was rooted using the closest BLAST hit for the *A. pisum* RlpA4 protein, as phylogenetic positioning of the *A. pisum* RlpA proteins has shown that identity of the bacterial source of aphid *rlpA* is unclear (13). Some distinct RlpA sequences were predicted to be derived from a single coding sequence—in these instances, the amino acid range is specified.
(TIF)

**S3 Fig. Bioinformatic workflow of aphid genome assembly and annotation.**
(TIF)

**S4 Fig. SDS-PAGE of protein fractions generated during affinity purification of recombinantly expressed, 6xHis-tagged *E. coli* AmiD.** Fractions analyzed include the insoluble pellet (P), soluble cell lysate (L), Ni-NTA flow-through (FT), lysis buffer wash (W1), HEPES buffer wash (W2), and HEPES-buffered elutions of increasing imidazole concentration: 50 mM (E1), 100 mM (E2), 200 mM (E3), and 500 mM (E4).
(TIF)

## Acknowledgments

We thank Eli Powell for assistance with aphid RNA extractions, Jerry Maeda for assistance with aphid embryo dissections, and the University of Texas at Austin Microscopy and Flow Cytometry Facility. We also thank Michael VanNieuwenhze for providing us with EDA.

## Author Contributions

**Conceptualization:** Thomas E. Smith, Nancy A. Moran.

**Data curation:** Thomas E. Smith, Yiyuan Li.

**Formal analysis:** Thomas E. Smith, Yiyuan Li.

**Funding acquisition:** Thomas E. Smith, Nancy A. Moran.

**Investigation:** Thomas E. Smith, Yiyuan Li, Julie Perreau.

**Methodology:** Thomas E. Smith, Yiyuan Li, Julie Perreau.

**Project administration:** Thomas E. Smith, Nancy A. Moran.

**Resources:** Thomas E. Smith.

**Software:** Thomas E. Smith, Yiyuan Li.

**Supervision:** Nancy A. Moran.

**Validation:** Thomas E. Smith.

**Visualization:** Thomas E. Smith, Yiyuan Li.

**Writing – original draft:** Thomas E. Smith.

**Writing – review & editing:** Thomas E. Smith, Yiyuan Li, Julie Perreau, Nancy A. Moran.

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
