## [Decision Letter · Decision Letter 0]

15 Nov 2021

Dear Dr Smith,

Thank you very much for submitting your Research Article entitled 'Comparative genomics of eight aphid subfamilies reveals variable relationships between host horizontally-transferred genes and symbiont peptidoglycan metabolism.' to PLOS Genetics.

The manuscript was fully evaluated at the editorial level and by independent peer reviewers. The reviewers appreciated the attention to an important problem, but raised some substantial concerns about the current manuscript. Based on the reviews, we will not be able to accept this version of the manuscript, but we would be willing to review a much-revised version. We cannot, of course, promise publication at that time.

Should you decide to revise the manuscript for further consideration here, your revisions should address the specific points made by each reviewer. Please take careful note of the critical comments from Reviewers #1 and #3 on the figures and revise them accordingly for clarity to help non-specialist and specialist readers. Also, it is important that you seriously address the criticisms of reviewer #3 regarding communicating the implications of your study. We will also require a detailed list of your responses to the review comments and a description of the changes you have made in the manuscript.

If you decide to revise the manuscript for further consideration at PLOS Genetics, please aim to resubmit within the next 60 days, unless it will take extra time to address the concerns of the reviewers, in which case we would appreciate an expected resubmission date by email to plosgenetics@plos.org.

[LINK]

We are sorry that we cannot be more positive about your manuscript at this stage. Please do not hesitate to contact us if you have any concerns or questions.

Yours sincerely,

Diarmaid Hughes

Associate Editor

PLOS Genetics

Kirsten Bomblies

Section Editor: Evolution

PLOS Genetics

Reviewer's Responses to Questions

**Comments to the Authors:**

Reviewer #1: I was very excited to read this paper. The addition of the genomes of aphid species from four new subfamilies marks an important contribution to the field, and I found the approach of analyzing gene presence of PGN-related genes in symbionts and host genomes compelling.

I really like the hypothesis that the authors put forward in which they propose that the presence of amiD allows the host to control the levels of Buchnera by regulating their PGN. However, my interpretation of the data differs slightly from that which the authors put forward. These are the major questions I have:

1. It seems unlikely that any of the Aphidini-associated Buchnera are able to produce PGN, given the lack of MurC, MurE and MurF. Rather, it seems like the PGN pathways in these endosymbionts are in the process of degradation – perhaps enabled by the loss of AmiD in the hosts. Have the authors checked whether the remaining genes are full length and not any pseudogenes?

2. Related to this: do the authors have any evidence for PGN in these organisms? Bacterial cell shape, mass spec/PGN probes, or expression of Aphidini? Or is there any in the literature? Same question for the Macrosiphini associated Buchnera. Has PGN been directly detected in these? If so this should be mentioned and if not this should be pointed out too.

3. It seems to me that these data show that ldcA is uncoupled from PGN synthesis in this system. The fact that Buchnera lack AmpG, AmpD and Mpl makes it difficult to imagine LdcA playing any role in this pathway and seems more likely to be carrying out an unrelated activity or perhaps interacting with a different host-associated bacterium (pathogens?).

4. If AmiD is regulating bacterial levels, does this mean that the Aphidini are less well controlled than the Macrosiphini? Is there an alternative way that the macrosiphini are controlled or is there some other difference in the relationship that might have facilitated this?

5. Overall, make it clearer (ideally in the figures) which Aphids have possible additional endosymbionts because this could affect the rationale behind the presence of the PGN genes in the hosts.

6. Did the authors check whether the genes being analysed were full length and not pseudogenes? If so, this should be mentioned.

Minor comments:

1. Figure 1 legend, could add Macrosiphini and Aphidini in brackets after the end of the first sentence Aphidinae species to help readers.

2. Fig 2 and 3 have been swapped – the legends don’t seem to match the figures as submitted. I believe Fig 3 is actually 2 and vice versa

3. Throughout, you can help readers who are not in the PGN field by being more explicit and adding more details. For example, in Fig 1 you could add FtsI in brackets next to PBP3. The authors use gene names in Fig 3 and protein names in Fig 1, which will be confusing to some readers. I would add more detail in brackets to nobody gets lost. For example, many of your readers may not be aware that mrdA is PBP2, mrdB is RodA

4. Similarly, in table S8 add common names PBP1/FtsI etc in addition to gene names. MrdB – rodA, mrdA – PBP2, mrcA, mrcB

5. SEDS proteins (RodA/FtsW) should be included in Fig 1 if you want to show all glycosyltransferase activity

6. I don’t have specific advice on how to fix this but I found Fig. 2 (called Fig. 3 – the one with gene gain/loss) very confusing to follow. I wonder whether there was any way the data could be presented more straightforwardly, perhaps by lining up the genes at the top of something. Once I figured everything out it was very comprehensive but I fear you may lose some readers with the complexity.

7. A specific note on this figure, the line to the right showing the superfamily Aphidoidea needs a line at end of Hormaphidinae to show that the others are from a different superfamily.

8. Fig 3 – you could help the reader by stating the superfamily along the top (even though they are all in the same). None aphid people will not be familiar with all of these so this will help readers move between Figs 2 and 3.

9. Fig 3, colored circles are unclear. Are they all supposed to be part of a single pathway? There are only pink circles and one black one. Not sure why these genes were selected.

10. line 293 – not sure this is really surprising, cell division can happen without PG eg chlamydia Ehrlichia. Could mention these.

Reviewer #2: Using comparative genomics, the authors revealed relationships between genes horizontally transfered from bacteria to aphids and the metabolism of peptidoglycan in their primary symbionts. The analysis is sound and thorough, with the obtained results interesting and important. I only have a few minor comments.

Lines 67-71: I understand what the authors mean. However, in many cases, Wolbachia, which is a main source of horizontally-acquired genes in host insects, is also categorized as a symbiont. I believe some modification is required.

Lines 77-79: This appears exaggeration and inconsistent with the description at lines 215-218.

Line 122, ~flagellar basal body (FBB) pathway: It would better to add a brief explanation here as to why this pathway was included in the analysis.

Lines 271, 369, 543, Candidatus S. alterna: I prefer "Skilesia".

Figures: It appears that figures 2 and 3 are oppositely placed.

Reviewer #3: The manuscript by Smith et al. reports the comparative analysis of pea aphid genomes and their Buchnera symbionts to investigate the distribution of genes involved in peptidoglycan (PG) metabolism, involved in host/symbiont relationships.

The authors analyzed 17 pair aphid and symbiont species, including four newly sequenced to increase the taxonomic coverage. The main objective was to understand "the rle of aphid horizontally-tranferred genes in symbiont cell wall metabolism".

The main results (resumed in the abstract and introduction) can be summarized as this:

(i) the PG gene repertoire varies for both host and symbiont; (ii)the distributing of three PG modeling gene families (rlpA1-5, amiD, ldcA) known to have been acquired by host pea aphids reveals that two (amiD, ldcA) co-occurr with symbiont PG genes, confirming that they are involved in PG modeling, whereas rlpA1-5 do not, indicating that they are not related to PG remodeling.

The authors conclude that future studies will be necessary to understand how these PG-remodeling acquired genes in aphids influence the symbiont PG metabolism.

I congratulate the authors for this interesting and thorough analysis, notably the sequencing of four additional aphid genomes. However, I had difficulties in appreciating the main implications, and I fear that these mainly descriptive results do not constitute a major advance for the subject, they are not conclusive on the function of the horizontally acquired genes other than what already known, nor is there experimental attempt at testing any arising hypothesis. As such, they may be suited for a more specialized audience than the large one of PloS Genetics.

Moreover, I found that the manuscript lacked in clarity and organization. Below some specific comments that may help improving it.

Figure 1 is very complete but also complex to look at. The authors may want to highlight the genes mentioned in the text (bold or red font for example).

My major concern if with Figure 2, which does not help understanding the results. There is no aphid tree in this figure, so it is impossible to follow the evolutionary events of loss and gain of PG genes. Figure 3 is not better, and I had a lot of trouble to go back and forth from it and the text.

The homologues were identified by simple BLAST searches, while more sensitive analysis (ex. HMM) may have been more suited, for example in the event of degenerated sequences or pseudogenization, a possibility not mentioned in the text.

All results described in pages 10-15 are not mentioned nor resumed in the abstract, and this is a pity. Even the title of the paragraph at line 208 does not make justice of all the work presented. Also, the hypothesis at the base of the analysis (that loss of PG remodeling enzymes in the aphid genomes may coincide with loss of PG synthesis in the symbiont, indicating functional linkage) seems trivial and may be better formulated.

Line 330, put "Discussion" in bold.

In conclusion, this is an interesting -although largely descriptive- analysis which may deserve a better structured text and clearer figures to highlight the novelty and impact of the results for the large audience.

**Have all data underlying the figures and results presented in the manuscript been provided?**

Reviewer #1: Yes

Reviewer #2: Yes

Reviewer #3: Yes

PLOS authors have the option to publish the peer review history of their article (what does this mean?). If published, this will include your full peer review and any attached files.

Reviewer #1: No

Reviewer #2: No

Reviewer #3: No

---

## [Decision Letter · Decision Letter 1]

7 Mar 2022

Dear Dr Smith,

Thank you very much for submitting your Research Article entitled 'Elucidation of host and symbiont contributions to peptidoglycan metabolism based on comparative genomics of eight aphid subfamilies and their Buchnera' to PLOS Genetics.

The manuscript was fully evaluated at the editorial level and by independent peer reviewers. The reviewers appreciated the attention to an important topic but reviewer #1 identified some concerns with the lack of an appropriate negative control in Fig. 5 that we ask you address in a revised manuscript. In addition, reviewers #1 and #3 have made suggestions to improve the clarity of the text that we suggest you implement.

We therefore ask you to modify the manuscript according to the review recommendations. Your revisions should address the specific points made by each reviewer.

[LINK]

Yours sincerely,

Diarmaid Hughes

Associate Editor

PLOS Genetics

Kirsten Bomblies

Section Editor: Evolution

PLOS Genetics

Reviewer's Responses to Questions

**Comments to the Authors:**

Reviewer #1: I am very happy to see the new EDA labelling data, this really strengthens the paper. However, I am concerned by a lack of appropriate negative control in Fig. 5. These probes are known to give high background labelling and whilst drug treatments are used to show specificity in Fig. 4 this uses a different labelling protocol in which bacteria are isolated from aphids. Therefore a suitable negative control is needed for Fig. 5 such as including PG targeting drugs during the last couple of hours of labelling or something similar.

The figure legend for Fig. 4 does not describe which strain is being used.

The labelling in figures 4 and 5 is unclear. Labelling microscopy panels as "AF488" or "cy5" is not helpful - the reader needs to know what is being labelled. So "EDA" or "PG label" is much more informative.

Reviewer #2: I believe this revised version is suitable for publication in PLOS Genetics.

Reviewer #3: The authors have answered my comments and the manuscript is now much clearer. They have also addded critical additional data that strengthens the impact of the work.

I still find figure 2 hard to follow but I understand you have done your best. I remain very skeptical on how species lacking murCEF can still produce PGN.

A few minor comments:

Abstract line 37: "The loss of amiD and ldcA HGTs coincides with symbiont PGN metabolism genes". This sentence is not very clear, i suggest to replace by "The loss of amiD and ldcA coincides with that of symbiont PGN metabolism genes".

Line 41: I was confused by you putting forward a hypothesis and then discarding it. Maybe add "The coincident loss of host amiD and symbiont murCEF in tribe Aphidini, in contrast to tribe Macrosiphini, may suggest...."

Line 61 "involvement in the synthesis of the symbiont cell wall"

Line 143: I would add: "In contrast with this hypothesis, we provide experimental evidence for ...."

**Have all data underlying the figures and results presented in the manuscript been provided?**

Reviewer #1: Yes

Reviewer #2: Yes

Reviewer #3: Yes

PLOS authors have the option to publish the peer review history of their article (what does this mean?). If published, this will include your full peer review and any attached files.

Reviewer #1: No

Reviewer #2: No

Reviewer #3: No

---

## [Editor Report · Decision Letter 2]

9 Apr 2022

Dear Dr Smith,

We are pleased to inform you that your manuscript entitled "Elucidation of host and symbiont contributions to peptidoglycan metabolism based on comparative genomics of eight aphid subfamilies and their Buchnera" has been editorially accepted for publication in PLOS Genetics. Congratulations!

Yours sincerely,

Diarmaid Hughes

Associate Editor

PLOS Genetics

Kirsten Bomblies

Section Editor: Evolution

PLOS Genetics

Comments from the reviewers (if applicable):

**Data Deposition**

http://datadryad.org/submit?journalID=pgenetics&manu=PGENETICS-D-21-01289R2

**Press Queries**

---

## [Editor Report · Acceptance letter]

2 May 2022

PGENETICS-D-21-01289R2 

Elucidation of host and symbiont contributions to peptidoglycan metabolism based on comparative genomics of eight aphid subfamilies and their Buchnera 

Dear Dr Smith, 

We are pleased to inform you that your manuscript entitled "Elucidation of host and symbiont contributions to peptidoglycan metabolism based on comparative genomics of eight aphid subfamilies and their Buchnera" has been formally accepted for publication in PLOS Genetics! Your manuscript is now with our production department and you will be notified of the publication date in due course.

With kind regards,

Livia Horvath

PLOS Genetics

On behalf of:
